# The IMpact of Vertical HIV infection on child and Adolescent SKeletal development in Harare, Zimbabwe (IMVASK Study): a protocol for a prospective cohort study

Ruramayi Rukuni [1,2] Celia Gregson,[3,4] Cynthia Kahari,[2,5] Farirayi Kowo,[6] Grace McHugh,[2] Shungu Munyati,[2] Hilda Mujuru,[7] Kate Ward,[8] Suzanne Filteau,[9] Andrea M Rehman [10] Rashida Ferrand[11]

For numbered affiliations see end of article.

**Correspondence to**
Dr Ruramayi Rukuni;
ruramayi.rukuni@lshtm.ac.uk

## ABSTRACT

**Introduction** The scale-up of antiretroviral therapy (ART) across sub-Saharan Africa (SSA) has reduced mortality so that increasing numbers of children with HIV (CWH) are surviving to adolescence. However, they experience a range of morbidities due to chronic HIV infection and its treatment. Impaired linear growth (stunting) is a common manifestation, affecting up to 50% of children. However, the effect of HIV on bone and muscle development during adolescent growth is not well characterised. Given the close link between pubertal timing and musculoskeletal development, any impairments in adolescence are likely to impact on future adult musculoskeletal health. We hypothesise that bone and muscle mass accrual in CWH is reduced, putting them at risk of reduced bone mineral density (BMD) and muscle function and increasing fracture risk. This study aims to determine the impact of HIV on BMD and muscle function in peripubertal children on ART in Zimbabwe.

**Methods and analysis** Children with (n=300) and without HIV (n=300), aged 8–16 years, established on ART, will be recruited into a frequency-matched prospective cohort study and compared. Musculoskeletal assessments including dual-energy X-ray absorptiometry, peripheral quantitative computed tomography, grip strength and standing long jump will be conducted at baseline and after 1 year. Linear regression will be used to estimate mean size-adjusted bone density and Z-scores by HIV status (ie, total-body less-head bone mineral content for lean mass adjusted for height and lumbar spine bone mineral apparent density. The prevalence of low size-adjusted BMD (ie, Z-scores <−2) will also be determined.

**Ethics and dissemination** Ethical approval for this study has been granted by the Medical Research Council of Zimbabwe and the London School of Hygiene and Tropical Medicine Ethics Committee. Baseline and longitudinal analyses will be published in peer-reviewed journals and disseminated to research communities.

## Strengths and limitations of this study

► This study will provide novel understanding of the effects of HIV on bone and muscle development in a large population of sub-Saharan African children living with HIV by using 'gold standard' size adjustment methods for dual-energy X-ray absorptiometry (DXA), which are crucial for assessing a population with inherent size differences.

► Bone architecture measurement using peripheral quantitative computed tomography (pQCT) will provide understanding of trabecular and cortical bone geometry and strength in children with HIV (CWH).

► This study will generate new data for total body and lumbar spine DXA, tibial pQCT, hand grip strength and standing long jump for Zimbabwean children without HIV which will inform normative reference data.

► While the age range in this study, 8–16 years, will allow analysis of pubertal delay in CWH, the follow-up period is insufficient to determine the impact on attainment of peak bone mass which probably occurs in the early 20s.

## INTRODUCTION

Sub-Saharan Africa (SSA) disproportionally bears the burden of global HIV infection, with nearly 90% of the estimated 2.1 million children under 15 years of age living in SSA.[1] The global scale-up of antiretroviral therapy (ART) has dramatically improved survival of children with HIV (CWH).[2] However, there is accumulating evidence that the growing number of these children are now reaching adolescence in SSA with multisystem chronic comorbidities associated with HIV infection and/or its treatment.[3]

Poor linear growth (ie, stunting) is one of the most common manifestations of perinatally (vertically) acquired HIV infection, affecting up to 50% of children.[4 5] Linear growth is greatest in adolescence during the pubertal development period. Bone mass is thought to change throughout the life course

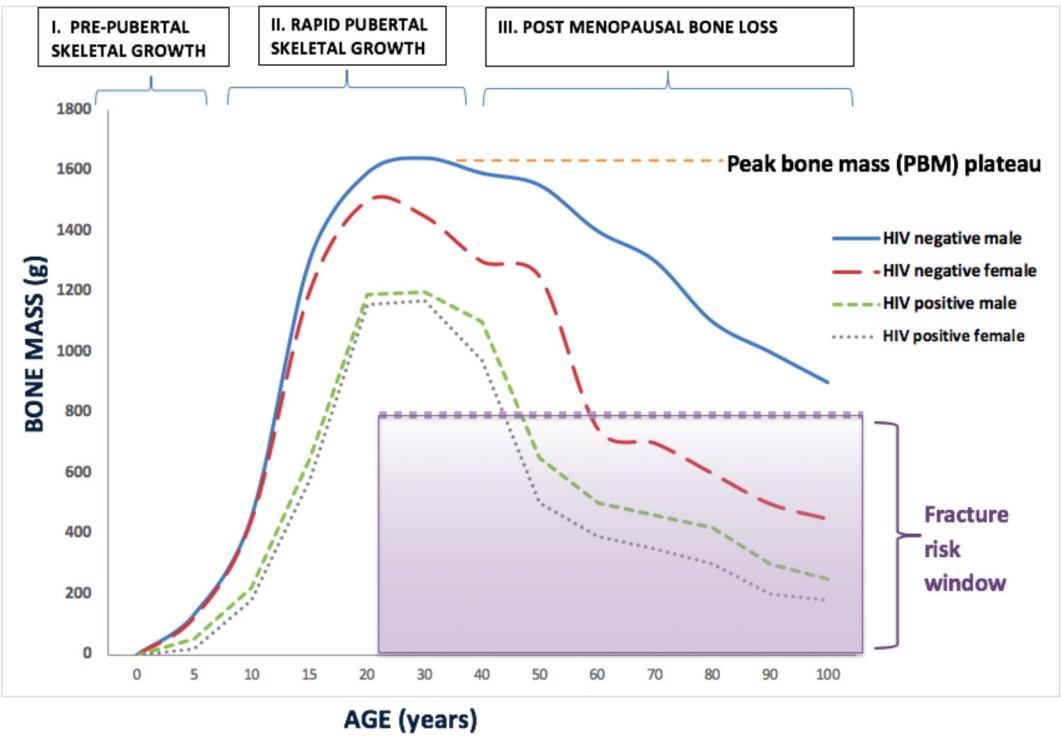

**Figure 1** Hypothesised changes in bone mass across the life course in HIV-infected and HIV-uninfected individuals.

and may be altered by HIV[6 7] (figure 1). The majority of peak bone mass (PBM), the maximum amount of bone accrued by the end of skeletal maturation, is attained during adolescent growth; by age 18 years in women and age 20 years in men, 80% of PBM is attained.[8] After PBM is reached, there is no net gain in bone mass. Therefore, PBM is the net reservoir of bone for later life, a key determinant of adult bone mineral density (BMD) and consequently of adult osteoporotic fracture risk.[9] Linear growth is therefore intimately linked to skeletal development but how HIV infection affects bone development in peripubertal SSA children is largely unknown. The prevalence of low BMD has been found to be higher in CWH than uninfected children in high-income and middle-income countries (7% in the USA,[10] 32% in Brazil[11] and 24% in Thailand[12]) compared with 1% in children without HIV in the USA.[10] No study has estimated the prevalence of low BMD in SSA, and the prevalence of and risk factors for low BMD in African CWH are not known.[6 13] It is important to highlight that the risk of poor bone accrual, reflected in low BMD measurements, is likely to be different in low-income countries compared with high-income countries due to factors such as malnutrition and social deprivation, but critically due to delayed ART initiation. A recent meta-analysis has shown that the median age of ART initiation in North America is 1 year, compared with 8 years in SSA.[14]

The mechanisms by which HIV may lead to low size-adjusted BMD in children are not fully understood but are likely multifactorial, including HIV-associated factors (eg, ART drugs, HIV disease stage) and traditional risk factors (eg, hypogonadism, smoking, alcohol, low physical activity and vitamin D deficiency).[15] HIV infection promotes systemic immune activation and production of inflammatory cytokines (eg, tumour necrosis factor α), which, in turn, promote increased bone resorption.[16] ART initiation, particularly with tenofovir (part of the first-line ART regimen in SSA), predicts an initial decline in BMD which stabilises after 2 years in adults.[17] It is thought that tenofovir may cause renal proximal tubule toxicity, resulting in phosphate wasting and increased bone turnover.[18] Although tenofovir and protease inhibitors have been associated with low BMD in adults,[19 20] studies in children have shown inconsistent findings.[21–23] Malnutrition, opportunistic infections and social deprivation may also impede musculoskeletal development. Reduced physical activity, associated with HIV,[24] may also impair muscle development and limit impact loading to reduce osteocyte-mediated bone accrual.[25 26] In adults, weak grip strength has been associated with increased falls and fracture risk.[27] Although muscle (lean) mass has also been shown to predict the magnitude of bone accrual during growth,[28] few studies have compared muscle strength and function between children with and without HIV. Interestingly, a small Canadian study showed deficits in muscle power in CWH.[29]

Another mechanism by which HIV may exert effects on BMD is through its effect on puberty. Even in the presence of ART, the onset of puberty is delayed by approximately a year in CWH in both high-income[30] and low-income settings.[31] Older age at ART initiation has been shown to be a significant risk factor for pubertal delay in Zimbabwean CWH.[31] Pubertal delay in HIV may be mediated through nutritional deficiency, recurrent infection or

chronic immune activation disrupting hormonal regulation.[31] Delayed puberty may be advantageous for linear growth; spending more time in puberty may allow more time for skeletal growth.[31] Conversely, delayed puberty has been shown in studies in high-income settings to be detrimental to bone mass accrual.[32 33] However, the impact of pubertal delay on BMD in low-income countries remains unknown. Pubertal delay can be assessed objectively using hand radiographs. Analysis of the growth plate development and fusion of long bones in the hands can accurately quantify bone age, which is a measure of skeletal maturation. Bone age lagging behind chronological age reflects pubertal delay.[34]

BMD is commonly measured by dual-energy X-ray absorptiometry (DXA) as two-dimensional (areal) BMD; however, this is highly dependent on bone size.[35] DXA underestimates areal bone density in short children, with smaller bones, and overestimates BMD in taller children, with bigger bones, despite the fact that they may have identical volumetric BMD. Size adjustment of DXA measures is therefore critically important in children with chronic diseases such as HIV, where smaller size due to poorer growth and delayed puberty may explain findings of lower BMD. The two 'gold standard' size-adjustment techniques chosen from the International Society for Clinical Densitometry are as follows: bone mineral apparent density at the lumbar spine (LS BMAD) and regression based total-body less-head (TBLH) bone mineral content (BMC) for lean mass adjusted for height (TBLH BMC$^{LBM}$)[36] Z-scores. As there are currently no published reference DXA data for child or adolescent populations in SSA, in this study we will use the best available data sets from high-income countries such as the UK[36] to generate Z-scores.

Unlike DXA, peripheral quantitative computed tomography (pQCT) takes into account bone size by directly measuring volumetric BMD. It has the additional advantage of separately assessing trabecular and cortical bone compartments, providing information on bone architecture. Furthermore, a range of bone strength indices, for example, strength stain index, validated against fracture risk can be calculated.[37 38] In high-income countries, markedly abnormal trabecular and cortical architecture have been shown in adults with HIV,[39] and abnormal bone architecture and impaired bone strength through to early adulthood have been shown in boys with HIV infection.[39] Few studies have assessed bone architecture and strength in CWH in SSA.

The IMVASK (IMpact of Vertical HIV infection on child and Adolescent Skeletal development in Harare, Zimbabwe) study aims to determine the prevalence of low size-adjusted BMD and muscle function (grip strength and standing long jump) in Zimbabwean children with and without HIV. pQCT assessment will enable understanding of the impact of HIV infection on bone architecture and strength. This study will further contribute to local reference data for DXA measures, bone age and

muscle function (grip strength and standing long jump) for an SSA population, establishing a biorepository for future research. Study results will aid understanding of bone and muscle accrual in the context of HIV infection in the era of ART.

## METHODS AND ANALYSIS
### Study objectives
This study aims to determine the impact of HIV infection on size-adjusted bone density in peripubertal children aged 8–16 years established on ART. In addition, this prospective study aims:
1. To quantify the prevalence of low size-adjusted BMD and low muscle function (grip strength and standing long jump) among CWH compared with uninfected children.
2. To investigate the risk factors for low size-adjusted bone density and low muscle function (grip strength and standing long jump) among CHW.
3. To compare the rates of bone mass accrual over 1 year between children with and without HIV and assess for interaction by pubertal stage to determine if CWH exhibit catch up growth.
4. To determine the differences in bone architecture measured by pQCT between children with and without HIV.

### Study hypothesis
We hypothesise that HIV infection adversely affects skeletal development, such that CWH, despite ART, accrue less bone mass and strength and have reduced muscle function during skeletal development.

### Study design
CWH aged 8–16 years and established on ART (n=300) and a comparison group of children without HIV, frequency-matched for age and sex (n=300) will be recruited into a prospective cohort study. Detailed musculoskeletal assessments will be conducted at baseline and after 1 year.

### Study setting
Parirenyatwa and Harare Hospital are the largest public-sector referral hospitals in Harare.[40 41] The paediatric HIV clinics at both hospitals provide HIV care to more than 2000 children. Although HIV care is increasingly decentralised to primary care level across the country, most children in Harare continue to receive care within HIV clinics in secondary healthcare facilities. Parirenyatwa hospital has a well-functioning radiology department which houses the University of Zimbabwe DXA and pQCT research unit and has access to private radiology services in the surrounding area. The hospital catchment areas have over 116 primary and 42 secondary government schools with an estimated 157 962 children enrolled.[42] School attendance in Harare province is high and

does not differ by HIV status, with 96% of children under 18 years attending school.[43]

## Recruitment of participants
### Eligibility
Inclusion criteria: age 8–16 years (includes prepubertal and peripubertal children), living in Harare, and in CWH only if:
i. Taking ART for at least 2 years (as adult studies demonstrate ART initiation is followed by an initial decline in BMD which stabilises after 2 years).[17]
ii. The child is aware of their HIV status, to avoid inadvertent disclosure as a result of study participation.

Children with perinatally acquired HIV will be included in this study. Perinatally acquired HIV will be defined based on Zimbabwean criteria, that is, self-report of no sexual debut or blood transfusions, a history of natural sibling or maternal HIV and characteristic clinical features of longstanding HIV. Children with horizontal infection will also be included in the study.

Exclusion criteria: acute illness (requiring immediate hospitalisation) and lack of consent.

Detailed information on all the above comorbidities will be collected using the main study questionnaire in the clinical history section. This information will be collected for both children with and without HIV. Comorbidities will not be used as the basis of excluding children from the study. However, for the purposes of deriving normative DXA data, those with severe bone disease will be excluded at the analysis stage.

### Recruitment of children with HIV
Systematic quota-based sampling by age and sex will be used to recruit 300 children from Parirenyatwa and Harare Hospital HIV clinics. Participants will be recruited sequentially as they attend clinic such that 50 male and 50 female will be chosen for each of three age strata: 8–10.99, 11–13.99 and 14–16.99 years. A maximum of five participants will be enrolled on each day for logistical reasons. The total number of children approached each day will be recorded, irrespective of whether they are subsequently eligible or enrolled to determine the sampling fraction. Written consent will be obtained from children and their guardians. Study processes and procedures will be clearly explained to children and their guardians and they will be given the option to accept or decline to take part in the research. It will also be explained that they are allowed to withdraw from the study at any time, for any reason, without affecting the care they receive from the clinic.

### Recruitment of children without HIV
Three hundred CWH will be randomly sampled from six government primary and secondary schools in the same catchment area as Parirenyatwa and Harare Hospitals. Younger children (8–12 years) will be selected from primary schools and older children (13–16 years) from secondary schools, with 13 year olds coming from both primary and secondary schools. The number of children selected from each school will be proportional to school size, thereby giving each child equal probability of being sampled. A random number sequence will be generated, and school registers will be used to select participants of similar age and sex as the children with HIV using the same quota-based approach of 50 male and 50 female in each of the three age strata. Guardians of selected school children will be invited to the study clinic to complete the consent process. Consenting participants will have a diagnostic HIV test as part of their assessment. Those testing HIV positive (anticipated to be approximately 2%–3%[44]) will be referred for HIV care.

## Study procedures
### Questionnaire
An interviewer-administered questionnaire together with hand-held medical records will be used to collect sociodemographic details and clinical history including age, sex, school attendance, orphan status, guardianship, history of fractures with mechanism of trauma, steroid use, smoking, alcohol, recreational drugs, family history of musculoskeletal disease, comorbidities, physical activity, diet and nutrition, and sun exposure. Where possible, validated instruments adapted for the local context will be used. For example, the International Physical Activity Questionnaire (IPAQ)[45] validated in multiple countries including South Africa and will be used to assess physical activity as multiples of the resting metabolic rate (MET) in MET-minutes. Diet and nutrition will be assessed using a tool we developed for the Zimbabwean context based on a validated dietary diversity and food frequency tool from India and Malawi[46] and international guidelines applicable to SSA.[47] The tool quantifies vitamin D supplementation and sunlight exposure and has been adapted to reflect the Zimbabwean context where fortification of oils and margarine with vitamin D is mandated by the government and specific vitamin D–rich foods such as kapenta fish are found.

### Clinical examination
A standardised musculoskeletal examination will be conducted using the validated paediatric Gait, Arms, Legs and Spine (pGALS) examination.[48] Additional clinical assessments will be carried out using standardised protocols and calibrated equipment. Anthropometry measurements will include standing and sitting height, arm span and mid-upper arm circumference. Height will be measured to the nearest 0.1 cm, by two separate readers using calibrated Seca 213 stadiometers. If the two height measurements differ by more than 0.5 cm, a third reading will be taken.[49] Weight will be measured to the nearest 0.1 kg using calibrated Seca 875 scales. Tanner pubertal staging will be carried out using a standardised protocol with an orchidometer to assess testicular volume in male participants.[50] Muscle function will be assessed in the upper and lower limbs by grip strength dynamometry and standing long jump, respectively. Hand grip strength

will be measured using a Jamar hydraulic hand-held dynamometer (Patterson Medical, UK) to the nearest 0.1 kg. Participants will be seated with the shoulder at 0° to 10°, the elbow at 90° of flexion and the forearm positioned neutrally. Three measurements will be taken from each hand in alternation and the highest measurement chosen. The standing long jump distance will be taken from the best of three correctly performed attempts to the nearest 0.1 cm, measuring the distance from the take-off line to the heel.

### Radiological assessments

DXA scans will be performed by two trained radiographers using a Hologic QDR Wi densitometer with Apex software V.4.5. Measurements will be taken from the lumbar spine, hip and total body. Fat and muscle mass will also be acquired; muscle mass is the fat-free mass measurement from DXA. DXA scans will be repeated in a subgroup (n=20) of participants to determine reproducibility. pQCT measurements of the non-dominant tibia will be taken using a Stratec XCT-2000 scanner (Stratec, Pforzheim, Germany) software V.6.20. Measurements of the non-dominant tibia will be taken at three sites at 4%, 38% and 66% of the tibial length, measured from the medial malleolus to the medial tibial plateau. Daily quality control will be performed by scanning the manufacturer-provided lumbar spine phantom for DXA and tibia phantom for pQCT. A radiograph of the non-dominant hand and wrist will be taken and used to quantify bone age using the Greulich and Pyle (G&P) atlas and the Tanner Whitehouse 3 (TW3) method. For intraobserver reliability, 10% of the radiographs will be randomly selected and rescored by the same operator after 1 week. For interobserver reliability, a different set of 10% of the radiographs will be rescored by a different expert. The estimated bone age will then be compared with the calculated chronological age.

### Blood tests

A fasting blood sample (up to 15 mL) will be collected. HIV markers (CD4 count and viral load) will be tested in CWH only. CD4 cell count will be measured using an Alere PIMA CD4 machine (Waltham, Massachusetts, USA). HIV viral load will be measured using the GeneXpert HIV-1 viral load platform (Cepheid, Sunnyvale, California, USA). The remaining blood plasma will be biobanked to enable future measurement of bone biochemistry. After removing the plasma, peripheral blood mononuclear cells will be isolated and cryopreserved. DNA will also be extracted using a manual method and stored for future genetic studies.

### Follow-up at 1 year

All study measurements, with the exception of DNA extraction, will be repeated after 1 year. Participants will be recalled exactly 1 year after their first DXA scan. The aim is to perform all scans within a 4-week window period. Contact will be maintained with participants

via regular phone calls and text messaging to minimise loss-to-follow-up. The schedule of study procedures is summarised in table 1.

### Outcome measures

The primary study outcomes are as follows:
1. Mean size-adjusted bone density Z-scores; TBLH BMC$^{LBM}$ and LS BMAD.[36]
2. The prevalence of low TBLH BMC$^{LBM}$ and LS BMAD Z-score<−2 at baseline.[36]

Secondary study outcomes are as follows:
1. Prevalence of low muscle function; grip strength and standing long jump-for-age (Z-score<−2) and musculoskeletal abnormalities/disabilities by HIV status at baseline.
2. Mean percentage change in TBLH BMC$^{LBM}$ (g) and LS BMAD (g/cm$^3$), tibial cortical and trabecular volumetric BMD (g/cm$^3$), total cross sectional area, cortical thickness and bone strength, muscle mass and function at baseline and 1 year, by HIV status.
3. Assessment of the extent to which pubertal delay explains changes in these bone and muscle outcomes.

### Sample size

The sample size was calculated to detect differences in DXA-measured mean size-adjusted bone BMD Z-scores between children with and without HIV. This study will have 80% power (α 0.05) to detect a 0.23 Z-score difference between 300 HIV-infected and 300 uninfected children, assuming an SD of 1.3. As there were no published studies from low income countries, estimates of the expected difference were taken from a US study of children with HIV aged 7–15 years.[10] In addition, our study will have 80% power to detect a 4.8% difference in the prevalence low size-adjusted BMD between the two groups. This is a smaller prevalence difference than that detected by the most conservative prevalence estimate of low BMD of 7% from three studies in high-income and middle-income countries.[10–12]

### Statistical analysis

For continuous variables that are normally distributed, the mean and SD will be presented. For skewed continuous variables, the median and IQR will be presented. Categorical variables will be summarised as frequencies and percentages. The distribution of demographic and clinical variables will be compared between CWH and without HIV using independent sample t-tests for means, Wilcoxon rank-sum test for medians and $\chi^2$ tests for proportions.

Baseline mean TBLH BMC$^{LBM}$ and LS BMAD Z-scores and the prevalence of low TBLH BMC$^{LBM}$ and LS BMAD Z-score will be compared between CWH and without HIV. Among CWH, the association between a priori defined risk factors (ART duration, ART type, proportion of life on treatment, age at ART initiation, CD4 count, viral load, bone age, pubertal stage, nutrition, socioeconomic status and orphanhood) against size-adjusted

**Table 1** Summary of study measurements to be quantified at baseline and follow-up

| | Measurement | Measurement method | Outcome |
|---|---|---|---|
| Interview-based questionnaire | Sociodemographic characteristics | Questionnaire | Age, sex, school attendance, orphanhood, guardianship |
| | Clinical history | Questionnaire* | History of fractures and trauma (modified Landin classification[55]) |
| | | | HIV history: age at diagnosis, WHO disease stage, nadir CD4 count, opportunistic infections† |
| | | | ART regimen/duration† |
| | | | Exposures: steroid use, smoking, alcohol, recreational drugs |
| | | | Family history of musculoskeletal disease and fractures |
| | | | Other comorbidities |
| | Physical activity | The International Physical Activity Questionnaire (IPAQ)[45] questionnaire (short form) | Median MET-minutes‡ of physical activity/week |
| | | | 1. Inactive (<600 MET-minutes/week) |
| | | | 2. Minimally active (600–1499 MET-minutes/week) |
| | | | 3. Highly active (≥1500 MET-minutes/week) |
| | Nutrition‡ | Dietary assessment tool (Modified Short Food Frequency Questionnaire[46])§ | Daily dietary calcium and vitamin D intake |
| | | | Prevalence of vitamin D supplementation |
| | | | Sun exposure |
| | Quality of life and disability | Washington Disability Score[56] | Functioning and disability score |
| Standardised examination | Musculoskeletal examination | Paediatric Gait, Arms, Legs and Spine (pGALS)[48] ±regional clinical examination | Joint, spine and gait abnormalities |
| | Pubertal stage | Tanner's staging[57 58] | Prepubertal (stage 1) |
| | | | Pubertal (stage 2–3) |
| | | | Postpubertal (stage 4 and 5) |
| | Anthropometry | Height (standing and sitting) | Standing height-for-age (Z-score)¶[59] |
| | | Weight | Weight-for-age (Z-score)¶[59] |
| | | Mid-upper arm circumference (MUAC)§ | Body mass index (BMI) (Z-score)¶[59] |
| | | | MUAC (Z-score)¶[59] |
| | Muscle strength | Jamar dynamometer | Hand grip strength (kg, Z-score)¶[60] |
| | | standing long jump¶ | Jumping distance (cm, Z-score)**[61] |
| Radiology | Skeletal maturity | Hand/wrist radiograph | Bone age (years) |
| | Bone and muscle composition | Dual-energy X-ray absorptiometry (DXA) of total body, lumbar spine and hip†† | Size corrected DXA measures of TBLH BMC^LBM (g), LS BMAD (g/cm³) and Z-scores <−2.¶ |
| | | | Lean mass |
| | Bone architecture | Peripheral quantitative computed tomography (pQCT) | Trabecular and cortical vBMD (g/cm³), total and cortical CSA (mm²), cortical thickness (mm), periosteal and endosteal circumference (mm), SSI (mm³), PMI (mm⁴) and CSMI (mm⁴) |
| Blood tests | Bone markers and DNA | Blood test (DNA extraction and serum saved) | Future testing e.g. Vitamin D, alkaline phosphatase, C-terminal telopeptide (CTX)‡‡ |
| | HIV markers | Blood test | CD4 count, HIV viral load† |

*Details of treatment and comorbidities will be confirmed by patient-held medical records where available.
†Denotes assessments to be carried out in HIV-infected participants only.
‡Energy requirements defined in METS (multiples of the resting metabolic rate that give a score in MET-minutes).
§Nutritional indicator to include composite information from history (usual diet last month, sun exposure–vitamin D status) and clinical examination (MUAC). Similar methods have been used in other low-income contexts.[46]
¶Age-specific and sex-specific Z-scores for (1) anthropometric measures will be determined using the WHO child growth standards.[59] (2) Hand grip strength will be determined with reference to the uninfected comparison group and European normative data.[60] (3) Jumping distance will be determined using normative data from South Africa.[61] (4) Low BMD will be determined with reference to published paediatric Hologic DXA reference databases for LS BMAD and TBLH BMC^LBM Z-scores.[36]
**Standing long jump; the longest distance after two attempts will be recorded.
††Pregnancy urine dipstick in female participants prior to DXA if uncertain pregnancy status.
‡‡Tests to be carried out on stored blood when further funding is secured.
CSA, cross-sectional area; CSMI, cross-sectional moment of inertia; DXA, dual-energy X-ray absorptiometry; LS BMAD, lumbar spine bone mineral apparent density; PMI, polar moment of inertia; SSI, Strength Strain Index; TBLH BMC^LBM, total-body less-head bone mineral content for lean mass adjusted for height.

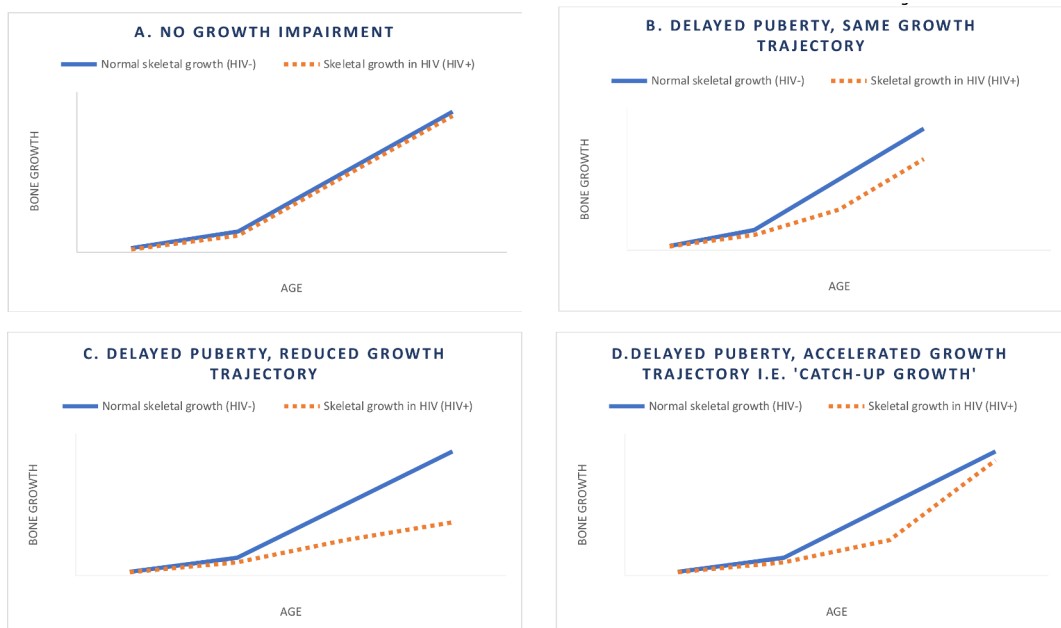

**Figure 2** Hypothesised growth scenarios to be assessed as interactions between pubertal stage and HIV status on change in bone mass.

BMD will be examined using multivariable linear regression (Z-score as a continuous variable) and multivariable logistic regression (as defined by the Z-score cut-off of $<-2$). Successful antiretroviral therapy will be defined as a viral load of less than 1000 copies/mL. Paired sample t-test or non-parametric Wilcoxon test will be used to assess for differences in TBLH BMC and LS BMAD on children with and without HIV between baseline and follow-up. Multivariable linear regression will be used to analyse the mean percentage change in TBLH BMC$^{LBM}$ (g) and LS BMAD (g/cm$^3$) between children with and without HIV. Models will be adjusted for physical activity, calcium and vitamin D intake. Interaction between the effects of pubertal stage (bone age) and HIV on change in TBLH BMC$^{LBM}$ and LS BMAD will be investigated to see if differences in bone density become more or less pronounced through puberty, that is, whether catch-up growth is possible, see figure 2. The regression coefficient (β) for percentage change in size-adjusted bone mass may suggest either no growth impairment (figure 2A), delayed puberty while maintaining the same growth trajectory (figure 2B) or delayed puberty with a reduced growth trajectory (figure 2C) in CWH. If β is markedly more positive in CWH, this suggests that catch-up growth may be possible (figure 2D). Pubertal delay in this study will be defined as the lack of the initial signs of puberty (Tanner stage 2) at an age that is more than 2 SD beyond the population mean[51] and as chronological age minus bone age >2 years.[52] Data for total body and lumbar spine DXA, tibial pQCT, hand grip strength and standing long jump in CWH will be analysed with reference to the comparator group of children without HIV.

For the purposes of normative data derivation, children without HIV who have any diagnosis or evidence of muscle or bone disease will be excluded. Then outliers with bone density, hand grip strength or standing long jump data beyond 2 SD from the mean will have their case record reviewed to exclude cases with underlying bone or muscle pathology. The remaining population will be used to generate normative references ranges for these quantitative traits.

### Data management

Data collection, management and storage will be governed by standard operating procedures and will follow the principles of Good Clinical Practice (GCP). Data will be captured using hand-held tablets for the questionnaires. Paper forms will be available in case of failure of electronic data entry. Microsoft Access will be used as the main back-end database as it allows programming of quality control checks and conditional data validation. GCP compliant audit trail modules will be incorporated into the databases and reports of aggregated data will be reviewed on a monthly basis. In order to assure data quality and consistency, all staff will receive regular training and regular quality checks will be conducted. Paper records will be stored for 8 years after the completion of research in secure, locked storage facilities. Field staff will download data to the central database, which is backed up onto an encrypted external hard drive daily, and to additional off-site and secure cloud back-up. The off-site back-up copies will be stored through the London School of Hygiene and Tropical Medicine (LSHTM) Research Data Management Support Service that has an established data repository. In order to preserve the long-term value of these data, it will be stored backed-up here indefinitely. Anonymised research data will be made available for sharing through the open access data repository

established by the LSHTM Data Management Support Service at the time of publication. This will allow other research groups to request access to study data and tools. Information on other researchers' data will be included in every study publication.

## Patient and public involvement

While patients were not directly involved in the design and conduct of the study, feedback from patient experiences in the study will be used to inform planned public engagement activities, which include science fairs, conducted by the research team at schools from where participants were recruited.

## Study status

Recruitment to this study began in May 2018 and is planned until August 2019. Study follow-up will run from May 2019 to August 2020.

## DISCUSSION

Although the scale-up of prevention of mother-to-child transmission has reduced perinatal HIV transmission, coverage is still not universal in most parts of SSA and therefore perinatal HIV infection is expected to affect large numbers of children for years to come. Furthermore, the scale-up of ART has reduced HIV-associated mortality dramatically so that CWH, who would previously have died in infancy or early childhood, are now reaching adolescence in increasing numbers. It is therefore important to understand the impact of HIV infection and its treatment on skeletal development during the critical period of puberty.

This study will determine the prevalence of low size-adjusted BMD in children with and without HIV in Zimbabwe, a country with a severe sustained early onset HIV epidemic. In addition, this study will determine risk factors for low size-adjusted BMD in CWH. We aim to identify factors amenable to intervention, which may be modifiable to maximise future bone health and minimise subsequent adult osteoporotic fracture risk. For example, reduced muscle function predicting low size-adjusted BMD may suggest targeted physiotherapy would be of benefit, which would warrant formal investigation.

Our study will provide insights regarding the mechanisms through which perinatal HIV infection affects the timing of pubertal onset and bone mass accrual. By measuring bone and muscle parameters at baseline and 1 year and by employing 'gold standard' size-adjustment methodology for DXA-measured BMD in the growing skeleton, this study will also provide insights into whether catch-up growth in terms of bone mass accrual is possible in HIV despite pubertal delay and provide age-related growth velocity data for CWH, with and without puberty.

While the age range in this study will allow analysis of pubertal delay in CWH, the follow-up period is insufficient to determine the impact on attainment of PBM, which probably occurs in the early 20s.[8] In addition, this study is not sufficiently powered to analyse the effect of individual ART types on size-adjusted bone density. An additional limitation is the inability to obtain accurate height data for CWH prior to enrolment to fully study growth recovery. This is problematic given the significant role of poverty and nutrition, independent of HIV status, in the first 1000 days of childhood[53]; this may explain some of the deficit in final height attained by CWH.

The bone architecture measured by pQCT in this study will provide separate assessments of trabecular and cortical bone density, and bone geometry and strength in Zimbabwean children. The evidence from studies in adult men established on ART demonstrates impairments in trabecular and cortical bone architecture.[54] Whether the same applies to children needs to be determined.

Furthermore, we will establish novel comparator data for DXA, pQCT, bone age, hand grip strength and standing long jump for a Zimbabwean population, which will be used for future research in this context. Although this represents the first steps towards developing normative reference data, the extent to which the children without HIV infection in this study are representative of the Zimbabwean population of 8–16 year olds is unknown. Furthermore, this study will establish a biorepository for future research, for example, potential bone turnover marker measurement and genotyping.

Given the magnitude of the HIV epidemic in SSA and the large cohort of young people who may experience impaired bone accrual, musculoskeletal disability or fracture as they reach adolescence and early adulthood, it is imperative to characterise the impact of perinatal HIV on musculoskeletal development.

## ETHICS AND DISSEMINATION

Ethical approval has been granted by the LSHTM Ethics Committee (Ref: 15333; 14 May 2018), the Institutional Review Board of the Biomedical Research and Training Institute (Ref: AP 145/2018; 20 February 2018), the Joint Research Ethics Committee for University of Zimbabwe College of Health Sciences and the Parirenyatwa Group of Hospitals (JREC) (Ref: 11/18; 1 March 2018), Harare Central Hospital Ethics Committee (HCHEC) (Ref: 170118/04; 23 February 2018), the Medical Research Council of Zimbabwe (Ref: MRCZ/A/2297; 10 April 2018) and the Ministry of Primary and Secondary Education Zimbabwe (Ref: C/426/Harare; 13 February 2018). This study is registered with the ISRCTN registry (Ref: ISRCTN12266984).

Study progress will be reported annually to MRCZ. Results of interim data analysis will be presented at national and international research meetings and conferences. Study findings will be published in international peer-reviewed scientific journals and disseminated to research communities at the end of study.

**Author affiliations**
[1]Clinical Research Department, London School of Hygiene and Tropical Medicine, London, UK
[2]Biomedical Research and Training Institute, Harare, Zimbabwe
[3]Musculoskeletal Research Unit, University of Bristol, Bristol, UK
[4]Older Person's Unit, Royal United Hospital NHS Trust, Bath, UK
[5]Department of Infectious Disease Epidemiology, Faculty of Epidemiology and Population Health, London School of Hygiene and Tropical Medicine, London, UK
[6]Department of Radiology, University of Zimbabwe, Harare, Zimbabwe
[7]Department of Paediatrics and Child Health, College of Health Sciences, University of Zimbabwe, Harare, Zimbabwe
[8]Lifecourse Epidemiology Unit, MRC, Southampton, UK
[9]Population Health, London School of Hygiene & Tropical Medicine, London, UK
[10]Infectious Disease Epidemiology, London School of Hygiene and Tropical Medicine, London, UK
[11]Clinical Research Department, London School of Hygiene and Tropical Medicine, London, UK

**Contributors** RR, RF and CG codesigned the study. RR wrote the study protocol and was responsible for journal selection and preparation of the first draft of this article as the principal author. CK contributed to the development of the peripheral quantitative computed tomography protocols. FK contributed to the development of the bone age analysis protocols. KW provided scan protocols, contributed to the study design, and gave methodological input regarding bone density size-adjustment and analysis. AMR contributed to the study design, in particular, sampling strategy, sample size calculation and the statistical analysis plan. SF provided advice regarding the development of nutritional assessment tools. GM, SM and HM advised on study conduct and provided study oversight. All authors reviewed and provided feedback on the manuscript prior to submission.

**Funding** This study is funded by the Wellcome Trust UK. RR is funded by Wellcome Trust UK grant number 206764/Z/17/Z. CK is funded by a National Institute of Health Fogarty Trent Fellowship (grant number 2D43TW009539-06). RF is funded by Wellcome Trust grant number 206316/Z/17/Z. Global challenges research funding from the University of Bristol established the sub-Saharan African MuSculOskeletal Network (SAMSON) enabling the provision of peripheral quantitative computed tomography in Zimbabwe for this study. AMR is additionally supported by the UK Medical Research Council (MRC) and the UK Department for International Development (DFID) under the MRC/DFID Concordat agreement which is also part of the EDCTP2 programme supported by the European Union grant reference (MR/R010161/1).

**Competing interests** None declared.

**Patient consent for publication** Not required.

**Provenance and peer review** Not commissioned; externally peer reviewed.

**ORCID iDs**
Ruramayi Rukuni http://orcid.org/0000-0002-2111-1311
Andrea M Rehman http://orcid.org/0000-0001-9967-5822

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
