## [Reviewer comments · BMJ Open]

ARTICLE DETAILS

TITLE (PROVISIONAL)	The IMPact of Vertical HIV infection on child and Adolescent Skeletal development in Harare, Zimbabwe (IMVASK Study): a protocol for a prospective cohort study
AUTHORS	Rukuni, Ruramayi; Gregson, Celia; Kahari, Cynthia; Kowo, Farirayi; McHugh, Grace; Munyati, Shungu; Mujuru, Hilda; Ward, Kate; Filteau, Suzanne; Rehman, Andrea M; Ferrand, Rashida

VERSION 1 – REVIEW

REVIEWER	Samsul Anwar Universitas Syiah Kuala, Indonesia
REVIEW RETURNED	14-Jun-2019

GENERAL COMMENTS	Dear Author(s), This research seems very significant in the field of global health especially in sub-Saharan Africa (SSA) regarding the impact of vertical HIV infection on child and adolescent skeletal development. Hopefully this study would lead to a new perspective of HIV and its impact on children and adolescent who received antiretroviral therapy (ART) so that we are able to minimize the risk of lower bone mineral density (BMD) and decrease of muscle function in Zimbabwe and elsewhere. Following the reviewing instruction of a study protocol, here are some points that I think the author(s) should pay close attention to. 1. Abstract on page 2 The author(s) did not write about the statistical methods going to be used to reach the objectives of the study. It has to be mentioned clearly in the abstract.2. Research ethics (e.g. participant consent, ethics approval) on page 7 Participant consent is not clearly defined for children with HIV (CWH). They should be given an option to agree or disagree to be involved in the research since there are some treatments they would receive as they become a participant of the study such as clinical examination, radiological assessments and blood tests. They should also be allowed cease their involvement anytime they want for any reason.3. Statistical methods used on page 10 Besides using linear regression methods (multiple linear regression and multivariate logistic regression), the author(s) could also use bivariate methods such as independent sample t test or nonparametric Mann-whitney test to check whether or not the main variables investigated (i.e. TBLH BMC and LS BMAD) were similar between CWH and children without HIV at the baseline and
--

	after one year. The author(s) could also check whether or not the time lapse (after one year) affected (reduce or increase) TBLH BMC and LS BMAD on CWH and children without HIV respectively using paired sample t test or nonparametric Wilcoxon test. In this case, I think this approach has advantages compared to linear regression analysis since the regression model and its interpretation depend on many aspects such as model assumptions and how fit the model is (i.e coefficient of determination). However, I understand that the author(s) used linear regression methods to investigate the relationship between the main variables and its risk factors (i.e. ART duration, ART type, proportion of life on treatment, age at ART initiation, CD4 count, viral load, bone age, pubertal stage, nutrition, socioeconomic status and orphanhood). I suggest the author(s) to use both approaches to conduct more comprehensive investigation of the problem studied. 4. References used on pages 15 to 18 In my opinion, the references used are appropriate. However, only 24 out of 60 (40.0%) references used were published under 5 years (≥ 2014) and only 37 out of 60 (61.7%) references used were published under 10 years (≥ 2009). The author(s) should add more recent studies to enrich their research background. Regardless of its shortcomings, I believe this study will give a significant contribution in understanding the impact of HIV on child and adolescent skeletal development in SSA. Thank you.
--	---

REVIEWER	Helena Rabie Stellenbosch University and Tygerberg Hospital South Africa I recruited adolescents for a co-hort study of adolescents with HIV CTAC
REVIEW RETURNED	22-Jun-2019

GENERAL COMMENTS	General comments Thank you for the opportunity to review this protocol currently enrolling HIV-infected children and adolescents in Harare, Zimbabwe. As the authors stated, moving beyond issues of survival and opportunistic infections requires documenting and quantification of long-term health concerns in HIV-infected children. This is essential to plan the additional care that will be required. This group of researchers have significant experience in working with adolescents and quantifying chronic health issues in HIV-infected children. The authors are proposing an assessment of, strength, height and bone health in children on antiretroviral therapy at baseline with a 12 month follow up period and also growth velocity in 1 year in. Importantly, adolescents and children will be matched with HIV-uninfected control children. This important long term issue that can not be studied in routine cohorts in low income settings because of the need for expertise and equipment that is not available in general settings and local guidelines do not mandate routine screening for these issues. Introduction
---

	Though puberty may be delayed in children with HIV as compared to those without it is important to add that a shorter time on antiretroviral therapy (age at initiation) is significant risk factor. Therefor mentioning recent data on age of initiation of therapy is important if it is available for the children in Zimbabwe. Methods section Study objectives  1) I assume the intention is to enrol children with vertically acquired HIV only. This needs to be stated. If this is not the intention state that horizontal HIV-infection will be included. 2) There is no mention of puberty delay in the objectives. Hypothesis.  1) Studying recovery of growth and bone health in children using anti-retroviral therapy is essential but a complex matter. 2) Poverty and food insecurity in early childhood plays a very significant role independent of HIV status. This may explain some of the gap in final height in children especially if they experienced poor nutrition in the 1000 days. It is likely that the study team will not have access to accurate data on the heights of children prior to enrolment on antiretroviral therapy and possibly even on therapy. This is a significant weakness of the study. 3) Including children who have access to antiretroviral therapy and are stable on therapy for 2 years and 8 years or older. It is not clear if the co-hort will represent majority intermediary and long term survivors and as such is different from children with early access to therapy. However children with therapy initiated soon after infection may have different therapy outcomes. Eligibility  1) Though an inclusive approach is essential to provide rich data some co-morbid conditions may possibly need to be excluded or very well documented and defined, this may contribute to severity of bone disease. This includes children with significant/ debilitation chronic lung disease, liver disease, renal disease or long term steroid use. It is mentioned that children with bone disease will be excluded, but only later. This should come earlier in the eligibility criteria. Also it needs to be discussed how these children will be managed and how they will be identified. 2) It should also be stated which control cases will be excluded and how these children will be screened apart from HIV tests. It would also be useful to define  1) Successful antiretroviral therapy 2) Pubertal delay – will this be pathological delay or a difference between infected and uninfected children only. The latter, though known to be present and significantly different but often a difference of months though statistically significant may be not necessarily be clinically meeting a diagnosis of delayed onset of puberty. 3) It is not stated what biochemistry of bone disease will be assessed. Ethical issues. Which clinical study register is used and what is the registration number. Discussion Apart from the clinical assessment and viral loads no parameters on mechanisms of bone disease is noted, only that it will be done thus comments on mechanism of any bone changes should be made with caution or it should be explicitly stated. Comments on catch up should also be made with caution as children will be followed up for a short period and many younger
--	--

	children will not start puberty. The authors state one of the strengths of this study is that it will provide data on the critical period of pubertal growth, this may in fact not be correct. Given the comment as noted, however a strength include that we will have age relate growth velocity data for children with HIV with and without puberty.
--	--

VERSION 1 – AUTHOR RESPONSE

RESPONSE TO REVIEWER 1

1. Abstract on page 2

The author(s) did not write about the statistical methods going to be used to reach the objectives of the study. It has to be mentioned clearly in the abstract.

The methods and analysis section of the abstract has been revised to include the statistical methods that will be used to assess the main study outcomes i.e. linear regression will be used to estimate absolute mean differences in size-adjusted bone density and Z-scores by HIV status.

2. Research ethics (e.g. participant consent, ethics approval) on page 7

Participant consent is not clearly defined for children with HIV (CWH). They should be given an option to agree or disagree to be involved in the research since there are some treatments they would receive as they become a participant of the study such as clinical examination, radiological assessments and blood tests. They should also be allowed cease their involvement anytime they want for any reason.

This section on recruitment of children with HIV has been written to define participant consent more clearly. It more clearly explains that participants are given an option to agree or disagree to be involved in the research since there are some treatments they would receive and that they are allowed to withdraw from the study at any time for any reason and that it would not affect the care they receive from the clinic.

3. Statistical methods used on page 10

Besides using linear regression methods (multiple linear regression and multivariate logistic regression), the author(s) could also use bivariate methods such as independent sample t test or nonparametric Mann-Whitney test to check whether or not the main variables investigated (i.e. TBLH BMC and LS BMAD) were similar between CWH and children without HIV at the baseline and after one year. The author(s) could also check whether or not the time lapse (after one year) affected (reduce or increase) TBLH BMC and LS BMAD on CWH and children without HIV respectively using paired sample t test or nonparametric Wilcoxon test. In this case, I think this approach has advantages compared to linear regression analysis since the regression model and its interpretation depend on many aspects such as model assumptions and how fit the model is (i.e. coefficient of determination). However, I understand that the author(s) used linear regression methods to investigate the relationship between the main variables and its risk factors (i.e. ART duration, ART type, proportion of life on treatment, age at ART initiation, CD4 count, viral load, bone age, pubertal stage, nutrition, socioeconomic status and orphanhood). I suggest the author(s) to use both approaches to conduct more comprehensive investigation of the problem studied.

Thank you for these suggestions; the section on statistical analysis has been revised to include the use of the suggested bivariate methods in addition to linear regression. The distribution of demographic and clinical variables will be compared between children with HIV and without HIV using t-tests for means, Wilcoxon rank sum test for medians and Chi-squared tests for proportions. Furthermore, the paired sample t test or nonparametric Wilcoxon test will be used to assess for differences in TBLH BMC and LS BMAD on CWH and children without HIV between baseline and follow up.

4. References used on pages 15 to 18

In my opinion, the references used are appropriate. However, only 24 out of 60 (40.0%) references used were published under 5 years (≥ 2014) and only 37 out of 60 (61.7%) references used were published under 10 years (≥ 2009). The author(s) should add more recent studies to enrich their research background.

Thank you for this comment. The literature review has been updated to include more recent references to enrich the background. Now only 12 of 56 (21%) references are from before 2014 and 4 out of 46 (7%) before 2009.

RESPONSES TO REVIEWER 2

1. Introduction

Though puberty may be delayed in children with HIV as compared to those without it is important to add that a shorter time on antiretroviral therapy (age at initiation) is significant risk factor. Therefore mentioning recent data on age of initiation of therapy is important if it is available for the children in Zimbabwe.

Thank you for highlighting this. The introduction has been revised to specify that age at ART initiation as a risk factor for pubertal delay as shown by a recent Zimbabwean study.

2. Methods section

Study objectives

- I assume the intention is to enrol children with vertically acquired HIV only. This needs to be stated. If this is not the intention state that horizontal HIV-infection will be included.

The section with the inclusion criteria has been updated to highlight that children with vertically acquired HIV will be enrolled into the study.

- There is no mention of puberty delay in the objectives.

The objectives have been updated to include pubertal delay more specifically. Object 3 has been expanded to describe how changes in bone mass accrual will be assessed in the context of pubertal stage between CWH and children without HIV.

3. Studying recovery of growth and bone health in children using anti-retroviral therapy is essential but a complex matter. Poverty and food insecurity in early childhood plays a very significant role independent of HIV status. This may explain some of the gap in final height in children especially if they experienced poor nutrition in the 1000 days. It is likely that the study team will not have access to accurate data on the heights of children prior to enrolment on antiretroviral therapy and possibly even on therapy. This is a significant weakness of the study.

This point has been included in the discussion as one of the limitations of the study.

4. Including children who have access to antiretroviral therapy and are stable on therapy for 2 years and 8 years or older. It is not clear if the cohort will represent majority intermediary and long term survivors and as such is different from children with early access to therapy. However children with therapy initiated soon after infection may have different therapy outcomes.

This cohort of children with HIV from the public hospital will largely represent children with vertical HIV. Of the cohort recruited so far, the earliest treatment was initiated was between 1 and 2 years of age, and the latest treatment was initiated was 13 years of age. The majority of children will have started on ART between age 6 and 10. We acknowledge that different therapy outcomes are hypothesised depending on when ART was initiated. ART duration and age will therefore be incorporated into causal inference models and adjusted for as required.

5. Eligibility

Though an inclusive approach is essential to provide rich data some co-morbid conditions may possibly need to be excluded or very well documented and defined, this may contribute to severity of

bone disease. This includes children with significant/debilitating chronic lung disease, liver disease, renal disease or long term steroid use. It is mentioned that children with bone disease will be excluded, but only later. This should come earlier in the eligibility criteria. Also it needs to be discussed how these children will be managed and how they will be identified. It should also be stated which control cases will be excluded and how these children will be screened apart from HIV tests. Detailed information on all the above comorbidities will be collected using the main study questionnaire in the clinical history section. This information will be collected for both children with and without HIV. Co-morbidities will not be used as the basis of excluding children from the study. However, for the purposes of deriving normative DXA data, those with severe bone disease will be excluded.

6. It would also be useful to define:

i. Successful antiretroviral therapy

In order to address the success of ART, viral load and CD4 testing will be carried out in CWH at baseline and follow up with undetectable viral load being the measure of successful therapy.

ii. Pubertal delay – will this be pathological delay or a difference between infected and uninfected children only. The latter, though known to be present and significantly different but often a difference of months though statistically significant may be not necessarily be clinically meeting a diagnosis of delayed onset of puberty.

A clinical definition of delayed puberty has been included in the manuscript. Delayed is defined clinically as the lack of the initial signs of puberty (Tanner stage 2 breast development in girls or testicular enlargement to ≥ 4 mL in boys) at an age that is 2–2.5 SDs beyond the population mean.

7. It is not stated what biochemistry of bone disease will be assessed.

Unfortunately, funding has yet to be secured to perform biochemical tests. We plan to apply for funding to test vitamin D, PTH, P1NP and CTX on stored plasma.

8. Ethical issues. Which clinical study register is used and what is the registration number.

This study is registered with the ISRCTN registry (Ref: ISRCTN12266984). This detail has been included in the ethics section.

9. Discussion

i. Apart from the clinical assessment and viral loads no parameters on mechanisms of bone disease is noted, only that it will be done thus comments on mechanism of any bone changes should be made with caution or it should be explicitly stated.

As stated above, funding has yet to be secured to perform biochemical tests. We plan to apply for funding to test vitamin D, PTH, P1NP and CTX on stored plasma. We intend to test the association between these bone markers of resorption and formation with bone and muscle outcomes to elucidate mechanistic components in the future.

ii. Comments on catch up should also be made with caution as children will be followed up for a short period and many younger children will not start puberty. The authors state one of the strengths of this study is that it will provide data on the critical period of pubertal growth, this may in fact not be correct. Given the comment as noted, however a strength include that we will have age related growth velocity data for children with HIV with and without puberty.

The sentence in the strength and limitations section that mentions that our study will provide data on the critical period of pubertal growth has removed. We have expanded the discussion to include that fact that our study will provide age related growth velocity data for CWH, in children through the spectrum of pubertal change on page 12.

VERSION 2 – REVIEW

REVIEWER	Samsul Anwar Universitas Syiah Kuala, Indonesia
REVIEW RETURNED	08-Nov-2019

GENERAL COMMENTS	Thank you to the authors for responding the points of my review on the first version of the manuscript. Responding to the author's answers, here are two minor points that I think need to be refined. Page 10 lines 38-39. The word “residuals” in the first sentence of the statistical analysis section should be removed since the residuals usually refer to the difference between the estimated value of the fitted model and the actual data. While for the descriptive purpose, it only needs to look at the data distribution instead of the residuals. Page 10 lines 43-44. The t-test in the first paragraph of statistical analysis section should be written as “independent sample t-test” since it compares different independent population (i.e CHW and without HIV).
--

VERSION 2 – AUTHOR RESPONSE

Reviewer Name: Samsul Anwar

Institution and Country: Universitas Syiah Kuala, Indonesia

Please state any competing interests or state ‘None declared’: None declared

Please leave your comments for the authors below:

Thank you to the authors for responding the points of my review on the first version of the manuscript. Responding to the author's answers, here are two minor points that I think need to be refined.

Page 10 lines 38-39.

The word “residuals” in the first sentence of the statistical analysis section should be removed since the residuals usually refer to the difference between the estimated value of the fitted model and the actual data. While for the descriptive purpose, it only needs to look at the data distribution instead of the residuals.

The word residuals has been removed from the manuscript.

Page 10 lines 43-44.

The t-test in the first paragraph of statistical analysis section should be written as “independent sample t-test” since it compares different independent population (i.e. CHW and without HIV).

This part of the manuscript has been rewritten as independent sample t-test.

VERSION 3 – REVIEW

REVIEWER	Samsul Anwar Universitas Syiah Kuala, Indonesia
REVIEW RETURNED	25-Dec-2019
GENERAL COMMENTS	From my perspective, the current analysis approaches are acceptable scientifically.